# The Effect of *Crataegus* Fruit Pre-Treatment and Preservation Methods on the Extractability of Aroma Compounds during Liqueur Production

**DOI:** 10.3390/molecules27051516

**Published:** 2022-02-23

**Authors:** Małgorzata Tabaszewska, Dorota Najgebauer-Lejko, Maria Zbylut-Górska

**Affiliations:** 1Department of Plant Product Technology and Nutrition Hygiene, Faculty of Food Technology, University of Agriculture in Krakow, Al. Mickiewicza 21, 31-120 Krakow, Poland; 2Department of Animal Product Technology, Faculty of Food Technology, University of Agriculture in Krakow, Al. Mickiewicza 21, 31-120 Krakow, Poland; dorota.najgebauer-lejko@urk.edu.pl; 3Department of Land Surveying, Faculty of Environmental Engineering and Land Surveying, University of Agriculture in Krakow, Al. Mickiewicza 21, 31-120 Krakow, Poland; maria.zbylut-gorska@urk.edu.pl

**Keywords:** hawthorn, volatile compounds, liqueur, PCA

## Abstract

The leaves, inflorescences, and fruits of hawthorn have long been known for their therapeutic properties. A wide range of hawthorn products, including liqueurs, are manufactured, due to the technological potential of the raw material as well as the richness of its volatile compounds. This study aimed to determine the effect of the liqueur production method and various methods of fruit preservation on the quantitative and qualitative composition of volatile compounds in the liqueurs produced. Hawthorn fruits saturated with sucrose and non-saturated with sucrose, fresh or preserved through one of three methods: freezing, air-drying, and freeze-drying, were used for liqueur preparation. The samples were analyzed using a gas chromatograph–mass spectrometer. They were found to contain 54 volatile compounds classified into 12 groups of chemicals. All 54 identified volatile compounds were detected in the liqueur made from hawthorn fruits non-saturated with sucrose and preserved by freeze-drying. In this liqueur type, 12 of the identified volatile compounds occurred in the highest concentration when compared to the other treatments. Among all volatiles, the following compounds were present in the analyzed liqueurs in the highest concentrations: dodecanoic acid ethyl ester (11.782 g/100 g), lactones (6.954 g/100 g), five monoterpenes (3.18 g/100 g), two aromatic hydrocarbons (1.293 g/100 g), isobensofuran (0.67 g/100 g), alcohol—2-methyl-2-propanol (0.059 g/100 g), and malonic ester (0.055 g/100 g). Among all analyzed liqueurs, the one made from the fruits non-saturated with sucrose and frozen was characterized by the smallest diversity of volatiles, which were present in the lowest concentrations in that liqueur.

## 1. Introduction

Hawthorn (*Crataegus*) belongs to the Rosaceae plant family. It is most common in Asia, North America, and Europe [1,2], as well as North Africa [3]. Its leaves, inflorescences, and fruits have long been known for their therapeutic properties [4]. Most often, however, these raw materials have been used to prepare preserves; jellies; jams; soft drinks, such as syrups; juices; alcoholic beverages, including liqueurs and wines; and sweets, including candies and fruits in syrup [1]. Hawthorn shrubs are also valued for their aesthetic qualities and are often planted in parks and gardens [5]. The most common of the 280 identified species of hawthorn include *C. monogyna*, *C. laevigata*, and *C. mexiana* [3].

A wide range of hawthorn products is manufactured due to the technological potential of the raw material, as well as the richness of its volatile compounds. The latter has been confirmed by Agiel et al. [6], who identified 54 volatiles in the inflorescences and immature fruits of *Crataegus azarolus* L. and *Crataegus pallasii* Grisb. The major compounds they found in the volatile essential oils of fresh and dried fruits included: tricosane, pentacosane, heptacosane, tetracosane, nonanal, undecanal, β-elemene, β-caryophyllene, eicosane, hexyl benzoate, benzoate, (E)-2-hexenyl benzoate, (E, E)-α-farnesene, and (Z)-3-hexenyl, and docasane. They also observed differences in volatile compound composition, which were affected by species examined and preservation method of the experimental material. They identified, i.a., (E)-β-damascenone only in the dried material, whereas δ-cadinene, germacrene D, α-selinene, β-selinene, δ-cadinol, selina-3,7(11)-diene, spathulenol, and valencene only in the fresh material from *C. azarolus*. Özderin et al. [4] detected as many as 81 volatiles in the flowers and leaves of hawthorn; the major ones included benzaldehyde, butyraldehyde, and (E)2-hexenal. Lakache et al. [7] identified 61 volatile compounds representing alkanes and alkenes, acids and esters, terpenes (mono-, di-, and sesquiterpenes), and aldehydes in hawthorn flowers and leaves.

Raw plant materials are usually seasonal and susceptible to fast decay. They need to be preserved to make them available all year round while their composition changes as little as possible [8]. For years, freezing has been the most popular method to ensure nutrient preservation. The best effects can be achieved via rapid freezing in liquid nitrogen (−80 °C) followed by gradient thawing −20 > ~−5 > ~4 °C [9]. Both the standard (slow) freezing to a temperature of −20 °C and further storage at this temperature trigger changes in the contents of active compounds [10,11]. Moreover, various drying methods are used to preserve plant materials. Of, these, freeze-drying has been claimed to be the most beneficial, due to its viability in preserving nutrients and aroma compounds, but, at the same time, it is the most expensive method. Therefore, air-drying is still in use, as it is relatively inexpensive, and because the dried material does not require any special storage conditions, unlike the freeze-dried one [8,12], even though it often modifies the composition and sensory traits of the dried material.

The beverage market, including alcoholic drinks, represents a dynamically developing industry branch. In order to succeed on this market, producers develop novel types of beverages and search for exceptional aromatic raw materials. Liqueurs, also called tinctures (*nalewki*), are one of the types of alcoholic beverages produced in Poland from wild-growing raw materials [13]. A liqueur is a sweetened spirit drink with a specified minimal sugar content depending on the type of aroma compound, produced by flavoring ethyl alcohol of agricultural origin and with the addition of products of agricultural origin or foodstuffs, with a minimum strength of 15% by volume [14]. The production and consumption of liqueurs has a longstanding tradition in Poland, and the interest in these alcoholic drinks has not been observed to wane [13].

Their quantitative and qualitative compositions in final products are affected by many factors, including the raw material preservation method and its saturation with a sucrose solution. The composition of volatile compounds in finished products is a driver of purchase decisions made by consumers and is essential in final product desirability assessment and consumption frequency. The available literature provides limited data on volatile compounds of products made from hawthorn fruits. There is no information either about compounds extracted from these fruits during liqueur production or about the effect of the fruit-preservation method and the drink-production method on the aroma values of the finished products. In view of the above, this study aims to determine the effect of the liqueur production method (involving the water content decrease in raw material using a sucrose solution for fruit saturation) and various methods of fruit preservation (involving low temperatures or water activity decrease through its evaporation using different techniques) on the changes in the quantitative and qualitative composition of volatile compounds in the liqueurs produced.

## 2. Results and Discussion

### 2.1. Volatile Compound Profile in Liqueurs from Accessory Hawthorn Fruits

Table 1 shows the statistically significant differences between particular types of liqueurs in respect of the content of identified volatile compounds with retention times and indexes and their classification into chemical groups along with their CAS numbers. Liqueurs produced from accessory hawthorn fruits differed significantly in terms of the nature and quantity of volatile compounds, which was affected by fruit pre-treatment and preservation methods (Table 1).

The analyzed liqueurs (Table 1) contained 54 volatile compounds classified into twelve groups of chemicals: acetals, alcohols, alhehydes, saturated and unsaturated hydrocarbons, aromatic hydrocarbons, esters, terpenes, and others. Of these, the most numerous were esters of fatty acids, except for the liqueur made from frozen hawthorn fruits non-saturated with sucrose (FnNS), in which acetals turned out to be the major group of volatiles (Figure 1). All identified volatile compounds were detected in the liqueur made from hawthorn fruits non-saturated with sucrose and preserved by freeze-drying (DNS). In this liqueur type, 12 of the identified volatile compounds occurred in the highest concentrations. The diversity of volatile compounds found in the liqueur made from hawthorn fruits non-saturated with sucrose but preserved by hot-air-drying was only slightly lower (53 volatile compounds; HNS). A similar number of volatiles (52) was identified in the liqueurs produced from hawthorn fruits saturated with sucrose and preserved by hot-air-drying (HS). The smallest diversity of volatiles (43 compounds) was found in the liqueurs made from fruits non-saturated with sucrose and preserved by freezing (FnNS). These liqueurs also featured the highest number of compounds with the lowest concentration of volatile compounds compared with all other analyzed liqueurs. 

Compared to the other analyzed liqueurs, the one produced from fresh hawthorn fruits non-saturated with sucrose (FNS) had statistically significantly higher concentrations of three out of four aldehydes identified in different treatments. In the liqueur produced from FnNS, eleven compounds present in other treatments were not detected at all, whereas concentrations of four compounds determined in these liqueurs were statistically significantly the lowest compared with the other liqueurs (Table 1). The liqueur produced from DNS hawthorn fruits contained eight volatile compounds (of which four were terpenes), with their concentrations being statistically significantly higher than in the other analyzed products. In the liqueurs produced from HNS fruits, statistically the highest concentrations were found for *p*-xylene and ethyl hexanoate, whereas *trans*-*p*-menth-8-en-2-one was not detected. In turn, the concentrations of six volatiles, including all alkenes, were statistically the highest, and concentrations of hexanol, *cis*-*p*-menth-8-en-2-one, and *trans*-*p*-menth-8-en-2-one were below the detection level in the liqueur made from fresh fruits saturated with sucrose (FS).

On the other hand, the liqueur made from hawthorn fruits saturated with sucrose and freeze-dried (DS) had statistically significantly the highest concentration of 2-methylbutanal but none of the volatiles identified in the other liqueurs, namely 1-butanol, 2-ethylhexanol, cis-*p*-menth-8-en-2-one, trans-*p*-menth-8-en-2-one, *p*-menth-6,8-dien-2-one, and methyl eugenol. The liqueur produced from HS of hawthorn fruits had statistically the highest concentrations of ethyl acetate, 9,12,15-octadecatrienoic acid, ethyl ester, and ethyl stearate but did not contain 1-butanol, *p*-menth-1-en-4-ol, cis-*p*-menth-8-en-2-one, and trans-*p*-menth-8-en-2-one. The saturation of accessory hawthorn fruits with a sucrose solution before preservation probably contributed to a greater distinctness of volatile compounds, because the concentration of only one compound (benzyl alcohol) was statistically the lowest in liqueurs produced from these fruits compared with the other analyzed drinks. However, the liqueurs made from fruits non-saturated with sucrose had a significantly higher number of volatile compounds occurring in higher concentrations compared with other liqueur types.

The pre-treatment (saturation with a sucrose solution) and preservation methods of hawthorn fruits contributed to changes in the concentrations of volatile compounds in particular groups of compounds (Figure 1). The method of hawthorn fruit preservation affected the contents of individual groups of volatile compounds, and its effect was more noticeable in the case of liqueurs made from fruits non-saturated with sucrose.

The liqueurs made from fresh fruits non-saturated with sucrose (Figure 1) had the highest concentrations of volatile compounds classified as esters and aldehydes and the lowest concentrations of those classified as acetals, alcohols, aromatic hydrocarbons, alkanes, and other volatiles. The liqueurs produced from frozen hawthorn fruits had the highest concentrations of acetals, aromatic hydrocarbons, and alkanes, whereas it had the lowest concentrations of esters and alkenes. In turn, the liqueurs prepared from air-dried hawthorn fruits had the highest concentrations of alcohols, alkenes, and other volatiles and the lowest concentrations of terpenes. Finally, the liqueur made from freeze-dried fruit had the highest concentration of terpenes and the lowest concentrations of other volatile compounds.

In the case of liqueurs made from fruits saturated with sucrose (Figure 1), the ratios of particular groups of volatile compounds were almost opposite compared to those made from the non-saturated hawthorn fruits. The lowest concentration of esters was detected in the liqueurs made from fresh fruit, whereas the highest was detected in those made from air-dried fruits saturated with sucrose. Likewise, the lowest concentrations of acetals and aromatic hydrocarbons were found in the liqueurs produced from fresh fruits, which additionally had the highest concentrations of the volatiles which were alcohols, alkanes, and alkenes. The liqueurs produced from frozen fruits had the lowest concentrations of terpenes and other volatile compounds, whereas the lowest concentrations of alcohols, alkanes, and alkenes were detected in the liqueurs made from air-dried fruits. In turn, the liqueurs prepared from freeze-dried fruits had the highest concentrations of aromatic hydrocarbons, terpenes, and other volatile compounds. 

Gasiński et al. [15], who analyzed beers made with the addition hawthorn fruits, identified 53 volatile compounds classified to nine chemical families. Similar to the present study, they belonged to alcohols, esters, ketones, monoterpenes, aromatic hydrocarbons, organic acids, aldehydes, and hydrocarbons. In this study, nine identified compounds were sesquiterpenes. Similar to the present study, the highest number of compounds identified by those authors came from the family of esters. Like the liqueurs analyzed in the present study, the beers with the addition of hawthorn fruits and juice from hawthorn fruits contained alcohols (1-hexanal), esters (hexanoic acid, ethyl ester, octanoic acid, ethyl ester, decanoic acid, ethyl ester, dodecanoic acid, ethyl ester), monoterpenes (limonene), and aromatic hydrocarbons (*p*-cymene). In turn, jams produced from *Crataegus azarolus* L. fruits were found to contain 56 volatile compounds [16], including *p*-cymene in concentrations similar to those found in the liqueurs produced from FnNS and DS hawthorn fruits. In the studied jams, the aforementioned authors also identified limonene. Its concentration was 50% higher than that determined in the liqueurs produced in our study from FnNS and DNS hawthorn fruits. Both the liqueurs made from hawthorn fruits and the hawthorn fruits analyzed by Agiel et al. [6] contained limonene, *p*-cymene, *p*-menth-1-en-4-ol, and *p*-menth-6,8-dien-2-one (carvone). Both in the present study and in the research by Agiel et al. [6], monoterpenes were not detected in all hawthorn species studied, all fruits harvested from various locations, and all fruits subjected to various treatment, as their content was below the detection limit. Zhao et al. [17] identified 61 volatile compounds in juices made from Chinese hawthorn fruits. They were classified to six groups of compounds, i.e., alcohols (16), esters (11), aldehydes and ketones (4), aromatic compounds (12), terpenes (8), furans (5), acids (4), and sulfur (dimethyl sulfide). Three of these identified alcohols were also detected in liqueurs produced from hawthorn fruits in our study (1-butanol, hexanol, 2-ethylhexanol), and their concentrations in the liqueurs were higher than in the aforementioned juices. Our liqueurs also contained two of the same esters as those found in the juices mentioned above (ethyl acetate, etyl hexanoate), and their concentrations were also higher in the liqueurs than in the juices. The amounts of volatile compounds determined in the analyzed liqueurs are similar despite their different concentrations and diversity affected by multiple factors.

Appendix A presents data on the general effect of sugar saturation and/or preservation method (least square means from the ANOVA) on the shaping of the profile of volatile compounds in the analyzed liqueurs [18,19,20,21]. Moreover, compounds in Appendix A were classified into groups, and descriptions of the flavors connected with particular compounds were provided according to data found in the literature. The saturation of the hawthorn fruits with a sucrose solution (Appendix A) had a statistically significant effect on a reduced content of volatile compounds from the group of acetals (acetaldehyde diethyl acetal, 2-methylbutyraldehyde diethyl acetal). Concentrations of other identified volatiles belonging to this group of compounds were statistically significantly higher in the liqueurs made from fruits saturated with a sucrose solution. The liqueurs produced from sucrose-saturated fruits had statistically significantly higher concentrations of 1-butanol and 2-ethylhexanol, whereas those made from the non-saturated fruits had statistically significantly higher contents of 2-methyl-2-propanol and benzyl alcohol. The saturation of hawthorn fruit with sucrose caused a statistically significant decrease in the concentrations of aldehydes identified in liqueurs, except for 2-methylbutanal, the concentration of which was statistically significantly higher. It also significantly reduced the concentration of 4-methyltetradecane representing alkanes. The concentration of alkenes in the liqueurs produced from hawthorn fruits non-saturated with a sucrose solution was significantly lower than in those made from saturated fruits. In contrast, the concentration of aromatic hydrocarbons was significantly higher in the liqueurs prepared from non-saturated fruits. The concentrations of esters of monounsaturated and polyunsaturated fatty acids were similar, regardless of fruit pre-treatment with sucrose, except for statistically significantly higher concentrations of butanedioic acid diethyl ester (Esters-MUFA) and 9,12,15-octadecatrienoic (Esters-PUFA) detected in the liqueurs produced from sucrose-saturated fruits. The concentrations of half of the identified esters of saturated fatty acids were statistically significantly higher in the liqueurs made from sucrose-saturated fruits, i.e., ethyl decanoate, ethyl dodecanoate, ethyl heptadecanoate, ethyl stearate, ethyl tetradecanoate, and ethyl undecanoate. In contrast, fruit pre-treatment with sucrose caused a significant decrease in the concentrations of the two following esters of saturated fatty acids: ethyl butyrate and ethyl hexanoate. Furthermore, this pre-treatment contributed to significantly lower concentrations of volatile compounds from the groups others and terpenes in the analyzed liqueurs.

In a study conducted by Vukoja et al. [22], the addition of disaccharides (sucrose, maltose, trehalose) during the manufacture of juice and cream from blackberries had a significant impact on in the concentrations of most identified volatile compounds within each chemical group of compounds, i.e., alcohols, acids, aldehydes and ketones, and terpenes. Sucrose addition significantly increased concentrations of the analyzed volatiles compared to other products made with the addition of other sugars [22]. Moreover, in cocoa products manufactured with various sugars (added in a total amount of 80%), the addition of sucrose contributed to increased concentrations of most identified volatile compounds compared to other added sugars, except for the concentration of aldehydes and ketones, which was similar [23]. Zlatic et al. [24] demonstrated that sucrose pre-treatment modified the concentrations of the identified volatiles proportionally to its addition level. Its 10% addition significantly increased the concentrations of, i.a., 1-butanol, 2-heptanol, 1-hexanol, 1-octanol, 2-phenylethyl alcohol, hexyl acetate, 2-methyl-buthyl butanoate, 2-methyl butanoate, benzaldehyde, 2-hexanal, *o*-cymene, α- and β-ionone, eugenol, 2-decanone, and geranic oxide. 

The liqueurs produced from fresh (non-preserved) hawthorn fruits had higher concentrations of 10 out of the 54 identified volatile compounds (3-methylbutyraldehyde diethyl acetal, 1-butanol, 2-ethylhexanol, benzyl alcohol, 3-methylbutanal, 2,4-dimethyl-1-heptene, 4-methyl-1-undecene, ethyl-9-hexadecenoate, ethyl octanoate, and 3-pinanone). The following flavor notes (defined from literature data) were perceptible in the analyzed liqueurs: fusel oil, sweet, balsam, whiskey, citrus, floral, chocolate, fruity wine, waxy, sweet, apricot, banana, brandy, pear, and cedar camphoraceous (Appendix A). The use of freeze-drying as the fruit-preservation method contributed to the increased concentrations of nine volatile compounds identified in liqueurs, i.e., ethylbenzene, *p- (o*-) cymene, diethyl methylsuccinate, ethyl decanoate, ethyl tetradecanoate, dill ether, 5,6-dihydro-2H-pyran-2-one, limonene, and *p*-menth-6,8-dien-2-one. These compounds, detected in the highest concentrations in liqueurs, significantly affected their flavor bouquets, by imparting the following flavor notes (Appendix A): ethereal, floral, sweet, citrus, solvent, sweet, waxy, fruity, apple, grape, oily, brandy, and minty licorice. In turn, the application of standard freezing for fruit preservation significantly increased the concentrations of five volatile compounds, namely 2-methylbutyraldehyde diethyl acetal, 4-methyltetradecane, styrene, ethyl butyrate, and acetophenone, compared with the liqueurs produced from the other fruits. These compounds (Appendix A) contributed to the more distinct balsamic, fruity juicy, fruit, pineapple, Cognac, almond, and flower notes of the liqueurs. In turn, hawthorn fruit preservation by air-drying caused a statistically significant increase in the concentrations of four volatile compounds identified in liqueurs, namely hexanol, *p*-xylene, ethyl hexanoate, and acetic acid. These compounds (Appendix A) contributed to the distinctly stronger aromatic sensations associated with the perception of herbal, plastic, cold meat fat-like, sweet, fruity, pineapple, waxy, green banana, and acidic notes. However, the most intensely perceived compounds were those which were detected in the highest concentrations in all liqueurs and, therefore, were the strongest determinants of their flavor notes (Appendix A), which included wax, ether, green nut, earthy, sweet, vegetable, fruity, creamy, and milky with a balsamic nuance. 

In another study, Sun et al. [25] identified 66 volatile compounds in a fresh pulp of hawthorn fruits, including 25.5% alcohols, 8.4% aldehydes, 4.6% ketones, 39.6% esters, 11.5% hydrocarbons, 0.3% furans, and 10.1% other compounds. In the freeze-dried pulp, they identified a lower number of volatile compounds, i.e., 54, including 27.7% alcohols, 13.9% aldehydes, 10.7% ketones, 34.2% esters, 5.5% hydrocarbons, 0.3% furans, and 7.7% others. The lowest number of volatiles—41—was detected in spray-dried pulp, including 42.5% alcohols, 25.4% aldehydes, 9.0% ketones, 14.4% esters, 4.4% hydrocarbons, 0.5% furans, and 3.8% other compounds. The above findings were inconsistent with the results of our study, where the highest numbers and concentrations of volatile compounds were found in the liqueurs made from dried fruits, regardless of the drying method employed. Likewise, in our study, the preservation technique applied influenced the composition of the individual groups of volatiles and, therefore, the development of flavor values of the liqueurs. Moreover, Agiel et al. [6], who analyzed the fresh and dried inflorescences and immature fruits of *Crataegus azarolus* L. and *Crataegus pallasii* L., noticed the effect of the fruit-preservation method on the concentration of volatile compounds in the analyzed samples. They detected (E)-β-damascenone only in the dried samples of *Crataegus azarolus* L., whereas α-selinene, β-selinene, δ-cadinene, germacrene D, selina-3,7(11)-diene, δ-cadinol, spathulenol, and valencene were only detected in the fresh samples of *Crataegus azarolus* L., and (2E,6E)-farnesol only in the dried samples, regardless of the hawthorn species.

### 2.2. PCA

The aim of the PCA was the additional verification of the obtained results. Multidimensional analysis provides a simple visual representation of the complex analysis of the collected data. This allows us to attempt to answer various questions, e.g., if there are relationships between the applied preservation method and the volatile compounds present in the sample or which preservation techniques provide similar results concerning the concentrations of volatile compounds. As a result of the information about the degree of correlation between the preservation method and the type and concentration of the volatile compounds, it is possible to select the best method of fruit preparation to achieve the final product, i.e., a liqueur with a rich, full-bodied, and pleasant flavor.

#### 2.2.1. Alcohols

Figure 2I presents a biplot summarizing the results of the principal component analysis performed for alcohols as variables. In the group of alcohols, the first two principal components (PC1 and PC2) together explained 72.4% of the total variance.

PC1 explains 51.84% of data variance for 1-butanol, hexanol, 2-ethylhexanol, benzyl alcohol, and 2-hexyloctanol, whereas PC2 explains 20.53% of data variance for 2-methyl-2-propanol and hexanol. The biplot analysis shows a high correlation between PC1 and 1-butanol, 2-ethylhexanol, and 2-hexyloctanol. The arrangement of the vectors of these variables indicates a positive correlation between them. What is more, a moderate correlation was noticed between hexanol and PC1 and a slightly weaker one between hexanol and PC2. In addition, positive correlations were noted among 1-butanol, 2-ethylhexanol, and benzyl alcohol, as well as between benzyl alcohol and 2-hexyloctanol. PC2 strongly correlated with 2-methyl-2-propanol. In the case of this compound, correlations with other variables were either weak or none (e.g., with 2-hexyloctanol). Moreover, a negative correlation was noticed between 2-hexyloctanol and hexanol.

The analysis of the distribution of cases relative to the variables allowed us to distinguish a few clusters of points that represented fruit-processing variants. The liqueurs made from the raw materials from the group A, including DNS, HNS, FnS, and DS, were characterized by a high concentration of 2-methyl-2-propanol (Figure 2). Those produced from FS had particularly high concentrations of 2-ethylhexanol and 1-butanol, and high concentrations of benzyl alcohol and 2-hexyloctanol. In turn, the liqueurs made from raw materials belonging to the group B (FnNS, HS) had higher hexanol contents. 

#### 2.2.2. Aldehydes

Figure 2II presents a biplot summarizing the results of the principal component analysis performed for aldehydes as variables. In the group of aldehydes, the first two principal components (PC1 and PC2) together explain 88% of the total variance. PC1 explains 61.10% of the data variance for 3-methylbutanal, benzylaldehyde, and 3-(2- or 4) methylbenzaldehyde), whereas PC2 explains 26.90% of the data variance for 2-methylbutanal. The biplot analysis demonstrates a strong correlation between PC1 and benzylaldehyde and 3-(2- or 4) methylbenzaldehyde, as well as a relatively strong correlation between PC1 and 3-methylbutanal. In addition, a strong correlation can be noticed between PC2 and 2-methylbutanal; a positive correlation between 3-methylbutanal, benzylaldehyde, and 3-(2- or 4) methylbenzaldehyde; and, finally, a lack of or very weak correlation between 3-methylbutanal and 2-methylbutanal and between benzylaldehyde and 2-methylbutanal. 

The analysis of the distribution of cases relative to the variables allowed us to distinguish a few clusters of points that represented fruit-processing variants. The liqueurs made from the FS, DS, and FnS fruits (group C) had a higher concentration of 2-methylbutanal. Liqueurs made from FNS had high concentrations of benzylaldehyde, 3-methylbutanal, and 3-(2- or 4-) methylbenzaldehyde. The liqueurs produced from the HNS, DNS, and HS fruits (group D) had low concentrations of all analyzed compounds, whereas liqueurs made from FnNS fruits had moderate concentrations of the analyzed compounds equal to the arithmetic means of all eight fruit-processing variants.

#### 2.2.3. Aromatic Hydrocarbons

Figure 2III presents a biplot showing the results of the principal component analysis performed for aromatic hydrocarbons as variables. In the group of aromatic hydrocarbons, the first two principal components (PC1 and PC2) together explain over 78% of the total variance. PC1 explains 50.08% of the data variance for toluene and *o*-xylene, whereas PC2 explains 28.63% of the data variance for ethylbenzene and *p*- (*o*-) cymene. The biplot analysis shows a relatively strong correlation between toluene and o-xylene and PC1, as well as a relatively strong correlation between PC2 and ethylbenzene and *p*- (*o*-) cymene. Positive correlations were demonstrated among toluene, o-xylene, and styrene, and among *p*- (*o*-) cymene, ethylbenzene, and *p*-xylene. A lack of or a very weak correlation was observed between styrene and *p*- (*o*-) cymene and between styrene and ethylbenzene. 

The analysis of the distribution of cases relative to the variables allowed us to distinguish a few clusters of points that represented fruit-processing variants. The liqueurs from a group E, produced from the FNS, FS, FnS, and HS fruits, had low concentrations of the analyzed compounds. The liqueurs produced from DS fruits had slightly higher concentrations of *p*- (*o*-) cymene and ethylbenzene, whereas those made from DNS fruits had very high concentrations of ethylbenzene and *p*- (*o*-) cymene compared with the liqueurs produced in the other processing variants. The liqueurs produced from the HNS and FnNS fruits (group F) had high concentrations of styrene, *o*-xylene, and toluene; however, the first also contained *p*-xylene, while the other contained significantly higher concentrations of toluene, *o*-xylene, and styrene. 

#### 2.2.4. Unsaturated Esters 

Figure 2IV presents a biplot summarizing the results of the principal component analysis performed for unsaturated esters as variables. In this group of compounds, the first two principal components (PC1 and PC2) together explain 80.80% of the total variance. PC1 explains 64.43% of the data variance for ethyl-9-hexadecenoate; 9,12-octadecadienoic acid, ethyl ester; 9,12,15-octadecatrienoic acid, ethyl ester; butanedioic acid diethyl ester; and diethyl malonate, whereas PC2 explains 16.37% of the data variance for diethyl methylsuccinate. The biplot analysis shows a strong correlation among butanedioic acid diethyl ester; ethyl-9-hexadecenoate; 9,12-octadecadienoic acid, ethyl ester; 9,12,15-octadecatrienoic acid; ethyl ester; and PC1, as well as a moderate correlation between PC1 and diethyl malonate. PC2 strongly correlated with diethyl methylsuccinate. A positive correlation can be observed among butanedioic acid diethyl ester; ethyl-9-hexadecenoate; 9,12-octadecadienoic acid; ethyl ester; 9,12,15-octadecatrienoic acid; and ethyl ester, whereas a negative correlation can be observed between ethyl-9-hexadecenoate and diethyl malonate. A lack of or a very weak correlation was noted between diethyl methylsuccinate and the following compounds: 9,12,15-octadecatrienoic acid, ethyl ester, and butanedioic acid diethyl ester.

The analysis of the distribution of cases relative to the variables allowed us to distinguish a few clusters of points that represented fruit-processing variants. Liqueurs from the group G, i.e., those made from the DNS and HNS fruits and especially those produced from DS fruits, had a high concentration of diethyl methylsuccinate. The liqueurs prepared from the FNS, FS and HS fruits (group H) had high concentrations of ethyl-9-hexadecenoate; 9,12-octadecadienoic acid, ethyl ester; 9,12,15-octadecatrienoic acid; ethyl ester; and butanedioic acid diethyl ester; however, the contents of these compounds were higher in liqueurs made from FNS and HS fruits than in those made from FS fruits. The liqueurs produced from FnS fruits had rather low concentrations of all analyzed compounds, whereas those produced from FnNS fruits had the highest concentration of diethyl malonate.

#### 2.2.5. Terpenes

Figure 2V presents a biplot summarizing the results of the principal component analysis performed for terpenes as variables. In the group of terpenes, the first two principal components together (PC1 and PC2) explain 91.6% of the total variance, with PC1 explaining 73.28% of the data variance for *p*-menth-1-en-4-ol, *cis-p*-menth-8-en-2-one, *trans-p*-menth-8-en-2-one, *p*-menth-6,8-dien-2-one, and limonene, and PC2 explaining 18.33% of the data variance for 3-pinanone. The biplot analysis demonstrates a very strong correlation between PC1 and *p*-menth-1-en-4-ol, *cis*-*p*-menth-8-en-2-one, *trans-p*-menth-8-en-2-one, and *p*-menth-6,8-dien-2-one, and a relatively strong correlation between PC1 and limonene. The arrangement of the vectors of these variables points to a positive correlation between them. In turn, PC2 correlated strongly with 3-pinanone, while there was no correlation or a very weak correlation between this compound and the other analyzed compounds.

The analysis of the distribution of cases relative to the variables allowed us to distinguish a few clusters of points that represented fruit-processing variants. Liqueurs made from the FS, DS, FnS, FnNS, HNS, and HS fruits (group I) had low concentrations of all analyzed compounds. The liqueurs produced from the FNS fruits had a high 3-pinanone concentration, whereas those made from the DNS fruits had especially high contents of *cis-p*-menth-8-en-2-one, *trans-p*-menth-8-en-2-one, *p*-menth-6,8-dien-2-one, and limonene.

#### 2.2.6. Others

Figure 2VI presents a biplot summarizing the results of the principal component analysis performed for the other volatile compounds as variables. In the group of other volatile compounds, the first two principal components together (PC1 and PC2) explain 75.5% of the total variance. PC1 explains 40.94% of the data variance for acetophenone, acetic acid, and methyl eugenol, whereas PC2 explains 34.53% of the data variance for dill ether and 5,6-dihydro-2H-pyran-2-one. The biplot analysis shows a strong and a relatively strong correlation between acetophenone, acetic acid, methyl eugenol, and PC1, as well as a strong correlation between dill ether and 5,6-dihydro-2H-pyran-2-one, and PC2, respectively. A positive correlation was also noticed between acetophenone, acetic acid, and methyl eugenol, and between dill ether and 5,6-dihydro-2H-pyran-2-one. In turn, no correlation or a very weak correlation occurred between 5,6-dihydro-2H-pyran-2-one and acetophenone and between acetophenone and dill ether.

The analysis of the distribution of cases relative to the variables allowed us to distinguish a few clusters of points that represented fruit-processing variants. The liqueurs from group J, i.e., those produced from the DNS and HNS fruits, had high concentrations of dill ether (especially those made from DNS) and 5,6-dihydro-2H-pyran-2-one. The liqueurs from group K, namely those made from the DS, HS, FS, FNS, and FnS fruits, had relatively low concentrations of the examined compounds. In turn, the liqueurs produced from FnNS fruits had high concentrations of acetophenone, acetic acid, and methyl eugenol.

## 3. Materials and Methods

### 3.1. Experimental Material

The experimental material included fruits (3500 g) of two-necked hawthorn (*Crataegus laevigata* (Poir.) DC.) collected (15 October 2015) at the consumption maturity stage on sunny day from a natural habitat located near a forest and far from asphalt roads and industrial buildings, in Jędrzejowo, Poland (50°38′22″ N 20°18′15″ E). The hawthorn grew on loamy, sandy–silty soil with a moderate humus content (1.6–1.75%) and a pH of 4.6–5.0. The soil had average contents of phosphorus—below 10 mg P_2_O_5_·100 g^−1^; potassium—15–20 mg K_2_O·100 g^−1^; magnesium—7–10 mg Mg·100 g^−1^; and mineral nitrogen—10–20 mg·kg^−1^ [26]. The following weather conditions were recorded in 2015 at the research station (Kielce) elevated 260 m above the sea level: total annual precipitation, 557 mm; insolation, 1973 h; average cloudiness, 5.4 octant; and average temperatures, 9.3 °C [27]. White sugar was purchased from Südzucker Polska S.A. (Wrocław, Poland), and ethanol 96% for liqueur preparation was purchased from Headquarters Seed Company Sp. z o. o. (Warszawa, Poland).

### 3.2. Liqueur Production Scheme 

After harvest, the raw material was cleaned of parts unsuitable for processing, washed under running water, and divided into 200 g portions to be used for further processing. At the first stage of the liqueur-production process, half the portion of accessory hawthorn fruits was saturated with sucrose by immersing the fruits in a 65% solution of sugar, under continuous stirring at 4 rpm, temperature of 65 ± 2 °C, for 35 min. In the subsequent stages, the above sucrose solution was added to impart desired sensory traits (taste) to the final product (Figure 3). The fruits non-saturated and saturated with sucrose were preserved with three methods (Table 2). The first method assumed the suppression of the microbial and enzymatic activities by fruit freezing, whereas the other two involved water-activity decrease by the removal of the involved water from the material using conventional hot-air-drying or freeze-drying. After they were prepared in such a way, the fruits were then subjected to the production procedures presented in detail in Figure 3.

Fruits pre-treated and preserved as described above were used to prepare liqueurs according to the scheme presented in Figure 3.

### 3.3. Methods

#### 3.3.1. Analysis of Volatile Compounds

The samples were analyzed by HS-SPME-GC/MS using a GCMS-QP 2010 Plus gas chromatograph–mass spectrometer (Shimadzu, Duisburg, Germany) and a 50/30 μm DVB/CAR/PDMS fiber (Supelco, Merck group, Poznań, Poland). Two GC columns (Phenomenex; Shim-Pol, Izabelin, Poland), one non-polar (Zebron ZB-5MSi 30 m × 0.25 mm, 0.25 µm) and the other polar (Zebron ZB-Wax 30 m × 0.25 mm, 0.25 µm), were used. Columns of different polarity were used separately (the analysis of the second column was used to confirm the identification of the compounds).

The quadrupole electron ionization (70 eV) mass spectrometer was operated in full scan mode in the range of 35–450 *m/z*; the temperature of the ion source was 250 °C. 

The parameters of the SPME autosampler Combi Pal System, AOC-5000 (Shimadzu, Duisburg, Germany) were set at: equilibration time: 30 min; temperature: 50 °C; exposition time: 15 min; temperature: 50 °C; and desorption in the splitless port: 240 °C for 2 min.

Chromatographic conditions were set at: flow (He, purity, 99.999; Linde Gaz Polska, Krakow, Poland): 1 mL·min^−1^, oven temperature: 37 °C (10 min), then increased to 132 °C (4 °C/min) and 240 °C (8 °C/min). The total time of the analysis was 60 min. 

The substances were identified by the mass spectra libraries NIST08, NIST08s, and FF NSC1.3, as well as by RI: retention index. RI data were retrieved from a reference database of the National Institute of Standards and Technology and compared—based on analyses of *n*-paraffins—with values calculated as Linear Retention Indices (LRI).

Results were expressed as the profiles of volatile compounds, i.e., percentage of the selected peak area in the sum of the peak areas of all the analyzed compounds.

#### 3.3.2. Statistical Analysis

All analyses were performed in triplicate (*n* = 3). The results obtained were subjected to statistical analysis using STATISTICA 13.3 software (Statsoft, Inc., Tulsa, OK, USA) and expressed as mean ± standard error of the mean. Either a one- or two-way analysis of variance was performed. The significance of the differences between the means was estimated based on Tukey’s post-hoc test at *p* < 0.05.

The effect of the fruit pre-treatment method on the content of volatile compounds in the finished product was investigated using the Principal Component Analysis (PCA). The PCA was used for those groups of volatiles whose individual components were most highly correlated. It was performed using STATISTICA 13.1 software (Statsoft, Inc., Tulsa, OK, USA).

## 4. Conclusions

The saturation of hawthorn fruits with a sucrose solution followed by their preservation using various methods caused changes in the qualitative and quantitative compositions of volatile compounds. In total, 54 volatile compounds were identified in the produced liqueurs. The liqueurs produced from fruits non-saturated with sucrose contained all of the identified volatiles.

The liqueurs produced from fruits preserved at low temperatures were characterized by the smallest diversities and the lowest concentrations of volatile compounds. Fruit preservation involving water-activity reduction (upon freeze-drying and air-drying) contributed to a more distinct flavor bouquet of the liqueurs (the greatest number and diversity of identified volatile compounds). Only liqueurs made from these fruits contained *cis-p*-menth-8-en-2-one and *trans-p*-menth-8-en-2-one.

Freeze-drying turned out to be the best method of hawthorn-fruit preservation applied to obtain liqueurs with the richest profile of volatile compounds. However, the traditional air-drying technique also provided good results in terms of the aromatic bouquet of the final products. On the other hand, the saturation of fruits with sucrose solution before their preservation did not produce a satisfying effect as regards the profile of flavor compounds. 

In the present work, the authors studied the effects of different types of fruit treatments on the profile of aroma compounds in hawthorn liqueurs. However, raw material was collected once from the restricted area, and liqueurs were produced in one series. Therefore, further research is needed to estimate the influence of different botanical origins of the raw materials on the selected characteristics of the final products. 

## Figures and Tables

**Figure 1 molecules-27-01516-f001:**
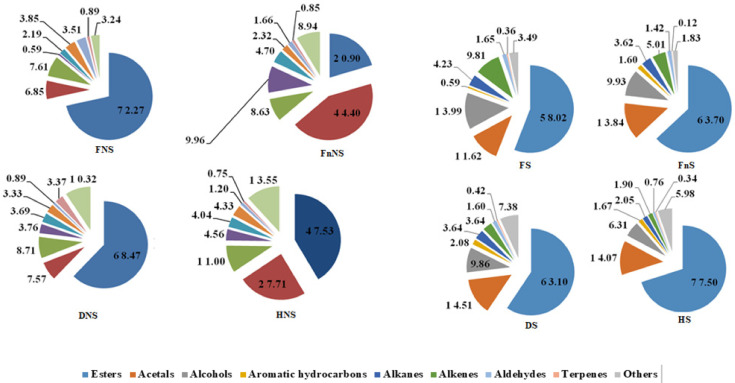
The content (%) of the respective groups of the volatile compounds in the liqueurs prepared from the hawthorn pseudo-fruits not-saturated with sucrose (NS) saturated with sucrose (S). Fresh (F); Frozen (Fn); Freeze-dried (D); Hot-air-dried (H).

**Figure 2 molecules-27-01516-f002:**
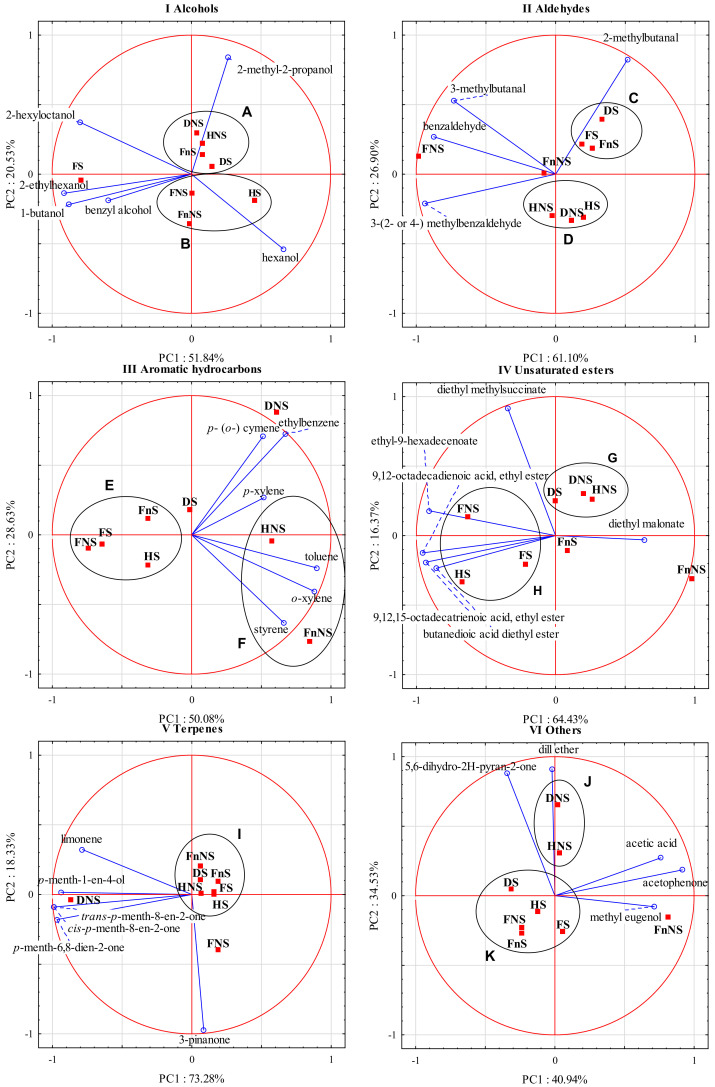
Principal components analysis of the volatile compounds (blue rings) of hawthorn liqueurs (red squares) (PC1 vs. PC2). (**I**) alcohols (groups A and B), (**II**) aldehydes (groups C and D), (**III**) aromatic hydrocarbons (groups E and F), (**IV**) unsaturated esters (groups G and H), (**V**) terpenes (group I), and (**VI**) others (groups J and K).

**Figure 3 molecules-27-01516-f003:**
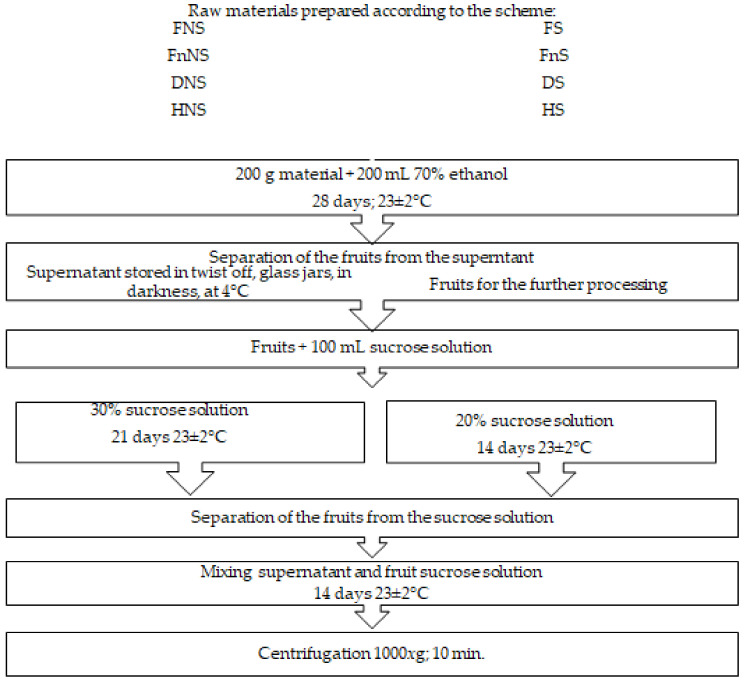
Scheme of liqueur production from pre-treated accessory hawthorn fruits.

**Table 1 molecules-27-01516-t001:** The content of identified volatile compounds (g·100 g^−1^) with their classification into chemical groups in the liqueurs prepared from hawthorn pseudo-fruits as affected by the sucrose saturation and preservation method (Mean ± SE).

Peak	R.t.	Name	ZB-5msi	ZB-Wax	CAS	Classification	Non-Saturated with Sucrose				Saturated with Sucrose			
			LRI	LRI			Fresh	Frozen	Freeze-Dried	Hot-Air-Dried	Fresh	Frozen	Freeze-Dried	Hot-Air-Dried
1	2.014	2-methyl-2-propanol	507	839	75-65-0	Alcohols	0.008 ± 0.000 a	nd	0.059 ± 0.001 d	0.048 ± 0.005 c	0.009 ± 0.000 a	0.027 ± 0.004 b	0.012 ± 0.001 a	0.024 ± 0.001 b
2	2.503	ethyl acetate	592	879	141-78-6	Esters	1.923 ± 0.061 a	3.154 ± 0.337 ab	5.127 ± 0.014 c	3.610 ± 0.317 b	1.883 ± 0.000 a	2.257 ± 0.243 ab	3.351 ± 0.013 ab	5.240 ± 0.169 c
3	2.979	3-methylbutanal	627	921	590-86-3	Aldehydes	1.368 ± 0.000 f	0.122 ± 0.002 b	0.033 ± 0.005 a	0.069 ± 0.006 a	0.693 ± 0.000 e	0.511 ± 0.025 d	0.247 ± 0.025 c	0.035 ± 0.000 a
4	3.073	2-methylbutanal	633	912	96-17-3	Aldehydes	nd	0.177 ± 0.123 bc	0.011 ± 0.001 a	0.044 ± 0.005 ab	0.283 ± 0.002 c	0.263 ± 0.007 c	0.502 ± 0.047 d	0.003 ± 0.000 a
5	3.153	acetic acid	638	1441	64-19-7	Acids	1.185 ± 0.017 ab	8.091 ± 0.328 d	2.552 ± 0.382 c	7.607 ± 0.238 d	1.113 ± 0.010 ab	0.670 ± 0.006 a	1.646 ± 0.236 b	2.558 ± 0.212 c
6	3.209	1-butanol	642	1156	71-36-3	Alcohols	0.061 ± 0.001 a	0.129 ± 0.018 b	0.082 ± 0.006 a	0.076 ± 0.038 a	0.360 ± 0.020 c	nd	nd	0.062 ± 0.001 a
7	4.667	acetaldehyde diethyl acetal	714	888	105-57-7	Acetals	1.544 ± 0.433 a	18.909 ± 1.790 d	2.532 ± 0.050 ab	19.479 ± 2.086 d	4.002 ± 0.288 ab	6.182 ± 0.497 b	5.305 ± 0.354 b	9.791 ± 1.029 c
8	5.812	Toluene	744	1020	108-88-3	Aromatic hydrocarbons	0.076 ± 0.000 a	0.550 ± 0.023 c	0.360 ± 0.012 bc	0.439 ± 0.077 c	0.154 ± 0.004 ab	0.161 ± 0.044 ab	0.383 ± 0.015 bc	0.338 ± 0.057 abc
9	7.717	ethyl butyrate	794	1023	105-54-4	Esters	0.131 ± 0.023 a	2.642 ± 0.272 b	0.407 ± 0.016 a	0.132 ± 0.029 a	0.089 ± 0.006 a	0.135 ± 0.025 a	0.238 ± 0.093 a	0.276 ± 0.016 a
10	10.276	2,4-dimethyl-1-heptene	833	873	19549-87-2	Unsaturated hydrocarbons	0.177 ± 0.047 a	0.205 ± 0.004 a	0.245 ± 0.034 a	0.061 ± 0.003 a	1.235 ± 0.159 c	0.674 ± 0.038 b	0.224 ± 0.010 a	0.073 ± 0.001 a
11	11.59	ethylbenzene	852	1103	100-41-4	Aromatic hydrocarbons	0.061 ± 0.034 a	0.169 ± 0.027 b	0.577 ± 0.042 d	0.269 ± 0.001 c	0.020 ± 0.000 a	0.173 ± 0.013 b	0.268 ± 0.014 c	0.050 ± 0.011 a
12	11.969	isobutyraldehyde diethyl acetal	857	975	1741-41-9	Acetals	0.440 ± 0.014 a	1.266 ± 0.256 b	0.251 ± 0.012 a	1.239 ± 0.214 b	1.883 ± 0.101 cd	1.562 ± 0.030 bc	2.132 ± 0.132 d	0.425 ± 0.143 a
13	12.156	*p*-xylene	860	1159	106-42-3	Aromatic hydrocarbons	0.267 ± 0.063 ab	0.357 ± 0.014 ab	0.852 ± 0.019 c	1.697 ± 0.174 d	0.153 ± 0.006 a	0.769 ± 0.056 c	0.450 ± 0.017 ab	0.524 ± 0.036 bc
14	12.792	Hexanol	869	1352	111-27-3	Alcohols	0.039 ± 0.014 a	0.121 ± 0.029 bc	0.021 ± 0.002 a	0.085 ± 0.004 b	nd	0.041 ± 0.003 a	0.040 ± 0.000 a	0.158 ± 0.011 c
15	13.607	Styrene	880	1237	100-42-5	Aromatic hydrocarbons	0.105 ± 0.020 a	7.297 ± 0.054 e	0.759 ± 0.034 c	0.975 ± 0.051 d	0.041 ± 0.002 a	0.062 ± 0.002 a	0.535 ± 0.137 b	0.506 ± 0.068b
16	13.734	*o*-xylene	882	1159	95-47-6	Aromatic hydrocarbons	0.062 ± 0.015 a	1.375 ± 0.172 d	0.500 ± 0.025 b	1.103 ± 0.113 c	0.070 ± 0.006 a	0.301 ± 0.019 ab	0.207 ± 0.063 a	0.215 ± 0.029 a
17	17.006	2-methylbutyraldehyde diethyl acetal	935	1060	3658-94-4	Acetals	1.471 ± 0.521 ab	22.372 ± 0.281 d	4.210 ± 0.209 c	4.781 ± 0.178 c	1.113 ± 0.359 a	1.732 ± 0.037 ab	2.210 ± 0.393 b	1.959 ± 0.009 ab
18	17.951	benzaldehyde	952	1499	100-52-7	Aldehydes	1.206 ± 0.020 c	0.888 ± 0.024 b	0.362 ± 0.003 a	0.458 ± 0.037 a	0.345 ± 0.018 a	0.434 ± 0.031 a	0.542 ± 0.029 a	0.378 ± 0.065 a
19	18.045	2-bromo-3-methylbutyraldehyde diethyl acetal	954	1064		Acetals	3.394 ± 0.540 c	1.858 ± 0.142 b	0.581 ± 0.017 a	2.212 ± 0.180 b	4.626 ± 0.086 d	4.359 ± 0.082 d	4.866 ± 0.217 d	1.891 ± 0.014 b
20	20.397	ethyl hexanoate	994	1221	123-66-0	Esters	0.295 ± 0.061 bc	0.227 ± 0.005 b	0.412 ± 0.000 cd	0.803 ± 0.037 e	0.070 ± 0.004 a	0.253 ± 0.023 b	0.214 ± 0.037 ab	0.530 ± 0.094 d
21	21.453	*p*- (*o*-) cymene	1015	1246	99-87-6	Aromatic hydrocarbons	0.017 ± 0.002 a	0.215 ± 0.010 bc	0.716 ± 0.119 d	0.077 ± 0.003 ab	0.156 ± 0.005 abc	0.138 ± 0.024 abc	0.275 ± 0.050 c	0.033 ± 0.001 a
22	21.689	Limonene	1020	1172	138-86-3	Monoterpenes	0.068 ± 0.001 a	0.813 ± 0.056 c	0.991 ± 0.178 c	0.453 ± 0.087 b	0.068 ± 0.001 a	0.067 ± 0.005 a	0.309 ± 0.018 ab	0.138 ± 0.017 a
23	21.915	2-ethylhexanol	1025	1485	104-76-7	Alcohols	0.150 ± 0.024 b	0.125 ± 0.029 b	0.032 ± 0.008 a	0.134 ± 0.002 b	0.445 ± 0.032 c	0.146 ± 0.006 b	nd	0.034 ± 0.007 a
24	22.112	benzyl alcohol	1029	1868	100-51-6	Alcohols	0.122 ± 0.003 c	0.151 ± 0.013 d	0.146 ± 0.007 cd	0.047 ± 0.003 b	0.136 ± 0.009 cd	0.051 ± 0.000 b	0.060 ± 0.009 b	0.023 ± 0.006 a
25	23.884	5,6-dihydro-2H-pyran-2-one	1066	1816	3393-45-1	Lactones	2.020 ± 0.218 ab	0.579 ± 0.011 a	6.954 ± 1.452 c	5.662 ± 0.678 c	2.214 ± 0.012 ab	1.060 ± 0.013 a	5.688 ± 0.712 c	3.366 ± 0.533 b
26	23.976	diethyl malonate	1068	1565	105-53-3	Esters	0.013 ± 0.001 a	0.055 ± 0.002 b	0.061 ± 0.001 b	0.041 ± 0.000 ab	0.052 ± 0.003 ab	0.023 ± 0.003 ab	0.027 ± 0.004 ab	0.026 ± 0.004 ab
27	24.155	3-(2- or 4-) methylbenzaldehyde	1071	1626	5973-71-7	Aldehydes	0.939 ± 0.016 d	0.477 ± 0.033 bc	0.484 ± 0.102 bc	0.626 ± 0.063 c	0.328 ± 0.012 ab	0.212 ± 0.007 a	0.309 ± 0.090 ab	0.349 ± 0.026 ab
28	24.263	2,4-dimethyl-1-decene	1074	1533	55170-80-4	Unsaturated hydrocarbons	2.302 ± 1.110 ab	1.104 ± 0.103 a	2.078 ± 0.195 ab	3.099 ± 0.131 b	6.213 ± 0.836 c	3.022 ± 0.042 b	2.238 ± 0.467 ab	1.135 ± 0.168 a
29	24.835	acetophenone	1085	1654	98-86-2	Phenyl methyl ketone	0.021 ± 0.004 ab	0.161 ± 0.011 d	0.077 ± 0.003 c	0.033 ± 0.001 b	0.016 ± 0.005 ab	0.009 ± 0.000 a	0.019 ± 0.007 ab	0.037 ± 0.008b
30	24.97	4-methyl-1-undecene	1088	1087	74630-39-0	Unsaturated hydrocarbons	1.370 ± 0.068 b	1.015 ± 0.068 ab	1.005 ± 0.243 ab	1.165 ± 0.204 ab	2.359 ± 0.350 c	1.312 ± 0.052 ab	1.180 ± 0.218 ab	0.693 ± 0.044 a
31	28.287	3-pinanone	1170	1544	18358-53-7	Monoterpenes	0.682 ± 0.032 e	0.041 ± 0.002 a	0.187 ± 0.041 cd	0.231 ± 0.002 d	0.134 ± 0.004 bc	0.045 ± 0.002 a	0.079 ± 0.004 ab	0.180 ± 0.014 cd
32	28.477	*p*-menth-1-en-4-ol	1175	1570	562-74-3	Monoterpenes	nd	nd	0.072 ± 0.000 e	0.003 ± 0.000 ab	0.017 ± 0.003 c	nd	0.029 ± 0.001 d	0.007 ± 0.000 b
33	28.677	butanedioic acid diethyl ester	1180	1676	123-25-1	Esters	0.077 ± 0.012 bc	nd	0.015 ± 0.003 a	0.028 ± 0.001 ab	0.091 ± 0.035 c	0.025 ± 0.004 ab	0.080 ± 0.009 bc	0.120 ± 0.017 c
34	28.801	dill ether	1183	1498	74410-10-9	Isobenzofurans	0.015 ± 0.002 a	nd	0.670 ± 0.180 b	0.235 ± 0.084 a	0.019 ± 0.004 a	0.078 ± 0.003 a	0.030 ± 0.001 a	0.003 ± 0.000 a
35	29.153	*cis*-*p*-menth-8-en-2-one	1192	1597	3792-53-8	Monoterpenes	nd	nd	0.452 ± 0.021 b	0.010 ± 0.001 a	nd	nd	nd	nd
36	29.326	ethyl octanoate	1197	1425	106-32-1	Esters	4.275 ± 0.185 c	0.338 ± 0.048 a	2.729 ± 0.567 b	0.595 ± 0.011 a	0.955 ± 0.013 a	3.780 ± 0.166 c	0.863 ± 0.114 a	0.884 ± 0.019 a
37	29.577	*trans*-*p*-menth-8-en-2-one	1203	1622	5948-04-9	Monoterpenes	nd	nd	0.278 ± 0.001	nd	nd	nd	nd	nd
38	29.746	diethyl methylsuccinate	1208	1743	4676-51-1	Esters	0.114 ± 0.001 b	nd	0.134 ± 0.009 b	0.129 ± 0.004 b	0.063 ± 0.002 a	0.052 ± 0.007 a	0.134 ± 0.033 b	0.062 ± 0.010 a
39	31.13	*p*-menth-6,8-dien-2-one	1246	1731	99-49-0	Monoterpenes	0.142 ± 0.019 bc	nd	1.387 ± 0.102 d	0.040 ± 0.001 abc	0.156 ± 0.004 c	0.012 ± 0.001 a	nd	0.019 ± 0.000 ab
40	31.184	5-propyldecane	1248	1261	17312-62-8	Saturated Hydrocarbons	1.995 ± 0.643 ab	2.630 ± 0.054 bc	2.881 ± 0.102 bc	2.727 ± 0.372 bc	3.430 ± 0.282 c	2.948 ± 0.008 bc	2.680 ± 0.226 bc	1.608 ± 0.064 a
41	33.51	2-hexyloctanol	1313	2141	19780-79-1	Alcohols	7.233 ± 1.620 ab	8.103 ± 0.270 abc	8.374 ± 1.155 abc	10.607 ± 1.250 cd	13.035 ± 0.450 d	9.666 ± 0.243 bc	9.750 ± 0.615 bc	6.006 ± 0.256 a
42	36.312	ethyl decanoate	1396	1631	110-38-3	Esters	1.095 ± 0.021 c	nd	1.766 ± 0.086 d	0.733 ± 0.011 a	0.922 ± 0.045 b	1.788 ± 0.002 d	1.062 ± 0.050 bc	1.107 ± 0.062 c
43	36.564	methyl eugenol	1406	1996	93-15-2	Methyl ether of eugenol Phenylpropanoids	nd	0.108 ± 0.000 c	0.063 ± 0.000 b	0.016 ± 0.002 a	0.131 ± 0.008 d	0.015 ± 0.001 a	nd	0.017 ± 0.003 a
44	38.011	4-methyltetradecane	1468	1450	25117-24-2	Saturated Hydrocarbon	0.195 ± 0.026 a	2.066 ± 0.217 e	0.810 ± 0.008 c	1.314 ± 0.115 d	0.804 ± 0.004 c	0.668 ± 0.008 bc	0.964 ± 0.026 c	0.445 ± 0.080 ab
45	38.658	ethyl undecanoate	1495	1743	627-90-7	Esters	0.061 ± 0.001 b	nd	0.150 ± 0.019 d	0.016 ± 0.001 a	0.151 ± 0.009 d	0.026 ± 0.007 ab	0.034 ± 0.007 ab	0.110 ± 0.019 c
46	40.71	ethyl dodecanoate	1611	1835	106-33-2	Esters	2.765 ± 0.090 a	6.122 ± 0.652 c	11.782 ± 0.452 e	4.475 ± 0.443 b	6.143 ± 0.187 c	10.059 ± 0.265 d	9.859 ± 0.341 d	10.706 ± 0.383 de
47	44.027	ethyl tetradecanoate	1795	2031	124-06-1	Esters	1.560 ± 0.128 bc	0.790 ± 0.113 a	1.860 ± 0.017 c	1.327 ± 0.027 b	3.726 ± 0.350 d	3.886 ± 0.117 de	4.706 ± 0.083 f	4.277 ± 0.016 ef
48	45.481	ethyl pentadecanoate	1910	2129	41114-00-5	Esters	0.666 ± 0.046 c	0.313 ± 0.042 a	0.560 ± 0.010 bc	0.415 ± 0.085 ab	0.504 ± 0.068 bc	0.513 ± 0.016 bc	0.517 ± 0.014 bc	0.653 ± 0.031 c
49	46.586	ethyl-9-hexadecenoate	1979	2267	68862-27-1	Esters	0.383 ± 0.041 b	nd	0.199 ± 0.005 ab	0.117 ± 0.037 a	0.525 ± 0.004 c	0.196 ± 0.001 ab	0.182 ± 0.002 ab	0.239 ± 0.051 b
50	46.851	ethyl hexadecanoate	1996	2239	628-97-7	Esters	47.409 ± 6.162 c	6.782 ± 0.282 a	35.704 ± 5.625 b	29.200 ± 0.501 b	32.741 ± 3.153 b	33.446 ± 0.793 b	36.047 ± 0.224 b	37.828 ± 2.529 bc
51	48.199	ethyl heptadecanoate	2099	2341	14010-23-2	Esters	0.206 ± 0.013 bc	0.049 ± 0.000 a	0.233 ± 0.002 cd	0.111 ± 0.019 ab	0.217 ± 0.005 cd	0.205 ± 0.003 bc	0.226 ± 0.022 cd	0.320 ± 0.078 d
52	49.275	9,12-octadecadienoic acid, ethyl ester	2166	2525	6114-21-2	Esters	4.315 ± 1.170 c	0.228 ± 0.013 a	2.756 ± 0.281 bc	2.023 ± 0.206 ab	3.676 ± 1.018 bc	2.769 ± 0.043 bc	1.932 ± 0.231 ab	5.014 ± 0.862 c
53	49.393	9,12,15-octadecatrienoic acid, ethyl ester	2173	2595	1191-41-9	Esters	5.946 ± 1.399 c	0.2000 ± 0.046 a	4.046 ± 0.113 bc	3.342 ± 0.111 bc	5.903 ± 1.451 c	3.746 ± 0.368 bc	3.0518 ± 0.377 b	8.731 ± 0.650 d
54	49.753	ethyl stearate	2195	2447	111-61-5	Esters	1.035 ± 0.160 b	nd	0.528 ± 0.030 a	0.438 ± 0.015 a	0.586 ± 0.091 a	0.542 ± 0.025 a	0.580 ± 0.014 a	1.379 ± 0.203 c

CAS: Chemical Abstracts Service, nd: not detected; different letters denote statistically significant differences between average values within a given compound at *p* < 0.05.

**Table 2 molecules-27-01516-t002:** List of equipment and parameters applied during preservation of the hawthorn fruits.

Freezing	
**Equipment**	**Description**
GP Gastro Group sp. z o. o sp. k., MBF8113GR, Lodz, Poland	Single layer of fruits was put on trays and placed in a freezing chamber at the temperature of −30 °C ± 1 °C for 120 h
**Freeze-Drying**	
Gamma 1–16 LSC, Christ (Germany) freeze dryer	In the first stage, the fruits were frozen on the trays to reach the temperature of −30 °C ± 1 °C using a one-chamber freezer (MBF8113GR, Poland) and were left at this temperature for 96 h. Freezing was performed in one layer (7.5 kg fruits per 1 m^2^). Freeze-drying was performed in a laboratory freeze dryer as follows: I stage (initial drying): temperature of the raw material: −30 °C; temperature of the condenser: −52 °C; shelf temperature: +20 °C; II stage—further drying (approx. 6 h): shelf temperature: +30 °C. The entire time needed to achieve assumed moisture content below 3% for the fruit input of 7.5 kg per 1 m^2^ was 24 h.
**Hot-Air-Drying**	
Food dehydrator Profi Line Hendi Food Service Equipment, TX Rhenen, Netherlands	Single layer of fruits was put on trays and placed in a drying chamber at the temperature of 45 °C ± 2 °C with forced air circulation for 120 h.

## Data Availability

The data presented in this study are available on request from the corresponding author.

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
