# Peer review of "The Effect of Crataegus Fruit Pre-Treatment and Preservation Methods on the Extractability of Aroma Compounds during Liqueur Production"

_molecules, 2022, doi:10.3390/molecules27051516_

Round 1

Reviewer 1 Report

The manuscript is well written and shows novelty to the reader. However, before publishing, I recommend some modifications in the abstract by adding quantitative data in abstract  sectionto interest the readers about the work

Author Response

Response to the Reviewer comments on manuscript ID: Molecules-1543539

Authors would like to thank for the positive response to our journal submission entitled:  „Effect of Crataegus fruit pre-treatment and preservation methods on extractability of aroma compounds during liqueur production”.  We appreciate the astute observations and constructive comments which helped us to improve our submission. The changes made to the document were marked up using Track Changes.

Below are our responses to Reviewer’s comments:

The manuscript is well written and shows novelty to the reader. However, before publishing, I recommend some modifications in the abstract by adding quantitative data in abstract  sectionto interest the readers about the work

The suggested by the reviewer changes were made in the manuscript, i.e. the following information was provided in the Abstract section of the manuscript:

Line 25-29: “Among all volatiles the following compounds were present in the analyzed liqueurs in the highest concentrations: dodecanoic acid ethyl ester (11.782 g/ 100 g), lactones (6.954 g/ 100 g), five monoterpenes (3.18 g/ 100 g), two aromatic hydrocarbons (1.293 g/ 100 g), isobensofuran (0.67 g/ 100 g), 28 alcohol - 2-methyl-2-propanol (0.059 g/ 100g) and malonic ester (0.055 g/ 100 g).”

Reviewer 2 Report

General comments

The authors present a comprehensive analysis of the effect of a number of preparation methods on the volatile profile of hawthorn fruits. The interest lies with the widespread use of hawthorn in a variety of products, though this paper deals specifically with the production of liqueurs. Generally, the design, analytical method and approach to data analysis are appropriate and the conclusions are supported by the results.

I do have a couple of significant issues that I would like the authors to address.

It is difficult to summarise data such as this so as to produce an understandable summary, and the authors have obviously made a lot of effort in this direction. The statistical analysis of the data was very thorough, comprising one-way and two-way analyses of variance intended to investigate the differences between the abundance of individual volatiles between the eight different treatments used; and the overall effects of sugar saturation and preservation respectively. I’m not sure that this really required two separate analyses. Table 2 was very difficult to understand, but it appears to represent the marginal means±SEs of the two-way analyses (this could be stated more clearly). I don’t see why the same overall result could not have been achieved by doing only the two-way analysis and forming linear contrasts between the relevant cells in that design.

The other arm of the analysis was the use of PCA biplots, which do not seem to contribute very much to the overall story. Scores plots are more commonly used in studies of quality or source attribution, but here the groups are predefined by the design and there are no unknown samples. Whether they cluster or not does not seem add anything to the overall picture presented in Tables 1 and 2. Similarly it is difficult to see what purpose the loadings data serves, unless correlations between volatiles could point to their biochemical origin, though the authors do not discuss this.

The extent of the analysis distracts attention from the fact that this is really quite a small dataset with each treatment being represented by three replicates. This in turn must significantly limit the reliability of the estimates of the mean and standard errors obtained and this should be acknowledged as part of the Discussion.

Minor points

The abbreviations list (line 29ff) is not complete.

Line 79 “liqueurs, also called liqueurs”?

Figure 1. Generally, very useful, but would benefit from using the same abbreviations as the rest of the paper. As it stands it is difficult to relate it directly to the tables or the text.

Line 241 (and in other places). Are the descriptions of the flavour notes based on actual taste tests, rather than the published descriptions of flavours/aromas? If so, this would represent a completely separate arm of the study and should be reported with a description of the methods used (however informal they might have been).

Figure 3. The scheme for liqueur production is not completely clear. In the third stage, the fruits seem to be subjected to two different treatments with sucrose solution. I assume that this related to the saturated/non-saturated arm of the study, but this should be made specific. Also, I don’t know what “intract” means, perhaps “supernatant”. It does not seem to be an English word.

Author Response

Response to the Reviewer comments on manuscript ID: Molecules-1543539

Authors would like to thank for the positive response to our journal submission entitled:  „Effect of Crataegus fruit pre-treatment and preservation methods on extractability of aroma compounds during liqueur production”.  We appreciate the astute observations and constructive comments which helped us to improve our submission. The changes made to the document were marked up using Track Changes.

Below are our responses to Reviewer’s comments:

The authors present a comprehensive analysis of the effect of a number of preparation methods on the volatile profile of hawthorn fruits. The interest lies with the widespread use of hawthorn in a variety of products, though this paper deals specifically with the production of liqueurs. Generally, the design, analytical method and approach to data analysis are appropriate and the conclusions are supported by the results.

I do have a couple of significant issues that I would like the authors to address.

It is difficult to summarise data such as this so as to produce an understandable summary, and the authors have obviously made a lot of effort in this direction. The statistical analysis of the data was very thorough, comprising one-way and two-way analyses of variance intended to investigate the differences between the abundance of individual volatiles between the eight different treatments used; and the overall effects of sugar saturation and preservation respectively. I’m not sure that this really required two separate analyses. Table 2 was very difficult to understand, but it appears to represent the marginal means±SEs of the two-way analyses (this could be stated more clearly). I don’t see why the same overall result could not have been achieved by doing only the two-way analysis and forming linear contrasts between the relevant cells in that design.

Response:

In Table 1 we wanted to show statistically significant differences between particular type of liqueurs in respect of the content of identified volatile compounds with retention times and indexes and their classification into chemical groups and their CAS numbers. On the other hand, Table 2 presents data on the general effect of sugar saturation and/or preservation method (lest square means from the ANOVA) on the shaping of the profile of volatile compounds in the analyzed liqueurs. Moreover compounds in Table 2 were classified into groups and description of the flavors connected with particular compounds was provided according to data found in literature. 

However, if the Reviewer or Editor consider that data in Table 2 presents data supplementary to Table 1, Table 2 can be moved to the Supplementary materials as Table S1.

Comment:

The other arm of the analysis was the use of PCA biplots, which do not seem to contribute very much to the overall story. Scores plots are more commonly used in studies of quality or source attribution, but here the groups are predefined by the design and there are no unknown samples. Whether they cluster or not does not seem add anything to the overall picture presented in Tables 1 and 2. Similarly it is difficult to see what purpose the loadings data serves, unless correlations between volatiles could point to their biochemical origin, though the authors do not discuss this.

Response:

The aim of PCA was additional verification of the obtained results. Multidimensional analysis provides a simple visual representation of the complex analysis of the collected data. This allows to attempt to answer a question e.g. if there are relations between the applied preservation method and volatile compounds present in the sample or which preservation techniques give similar results as regards concentration of volatile compounds. Thanks to the information as to degree of correlation between preservation method and kind and concentration of the volatile compounds it is possible to select the best method of fruit preparation to achieve final product, i.e. liqueur with rich, full-bodied and pleasant flavor.

Comment:

The extent of the analysis distracts attention from the fact that this is really quite a small dataset with each treatment being represented by three replicates. This in turn must significantly limit the reliability of the estimates of the mean and standard errors obtained and this should be acknowledged as part of the Discussion.

 Response:

The authors agree that there are possible significant differences between the results obtained during different series of production. In general, different results may be also obtained for plant material derived from different geographical area, during different time or in dependence on weather or soil conditions. Therefore, to minimize variability factors we use one lot of raw material. The aim of the study was to estimate the effect of sugar saturation and method of preservation on the number and concentration of aroma compounds in liqueurs. All analyzed treatments were produced from the same raw material to minimize the effect of other factors on the studied features.

However, the authors decided to provide additional information as regards the limitation of the study to the Conclusion section, i.e.:

In the present work, the authors studied the effect of different types of fruit treatments on the profile of aroma compounds in hawthorn liqueurs. However, raw material was collected once from the restricted area, also liqueurs were produced in a one series. Therefore, further research is needed to estimate the influence of different botanical origin of the raw materials on the selected characteristics of the final products.  (Page 19, lines 38-42)

Comment:

Minor points

The abbreviations list (line 29ff) is not complete.

The suggested by the reviewer changes were made in the manuscript, i.e.:

Line 41-48: “PC – principial components

PCA - principal components analysis

CAS - chemical abstracts service

PUFA – polyunsaturated fatty acids

MUFA – monounsaturated fatty acids

SAFA - saturated fatty acids

HS-SPME-GC/MS- headspace solid-phase microextraction- gas chromatography–mass spectrometry

Line 79 “liqueurs, also called liqueurs”?

The suggested by the reviewer changes were made in the manuscript, i.e.:

Line 96: “Liqueurs, also called tinctures (nalewki), are one of the types(…)”

Figure 1. Generally, very useful, but would benefit from using the same abbreviations as the rest of the paper. As it stands it is difficult to relate it directly to the tables or the text.

Response:

The abbreviations in Fig. 1 were amended

Comment:

Line 241 (and in other places). Are the descriptions of the flavour notes based on actual taste tests, rather than the published descriptions of flavours/aromas? If so, this would represent a completely separate arm of the study and should be reported with a description of the methods used (however informal they might have been).

Response:

Flavor compounds were described on the basis of published data found in literature: these data were also presented in Table S1.The change was made in the text, i.e.:

The following flavor notes (defined from literature data ) were perceptible in the analyzed liqueurs: fusel oil, sweet, balsam, whiskey, citrus, floral, chocolate, fruity wine, waxy, sweet, apricot, banana, brandy, pear and cedar camphoraceous (Table S1).” (Page 11, lines 44-47)

Comment:

Figure 3. The scheme for liqueur production is not completely clear. In the third stage, the fruits seem to be subjected to two different treatments with sucrose solution. I assume that this related to the saturated/non-saturated arm of the study, but this should be made specific. Also, I don’t know what “intract” means, perhaps “supernatant”. It does not seem to be an English word.

Response:

The legend to Figure 3 was improved and additional information given in the M&M section, Page 16, lines 40-41 to clarify the procedure of fruit pre-treatment and production of liqueur. Generally, Figure 3 present scheme of liqueur production from pre-treated fruits and the procedures for fruit pre-treatment (saturation with sucrose, preservation) were given in the text (and not included in the Figure 3).

Reviewer 3 Report

In addition to the comments in the pdf, I would urge to authors to address the following issues:

  • lack of technical (processing) repeats. As with any product that requires processing, this part can be a big source of variation
  • how were the compounds identified (there is a footnote for Table 1 that belongs in the M&M section) and how were they quantified? This is also relevant for the stats
  • The presentation of the results is exhaustive, please rethink it to make the reader follow the thinking and not simply repeat the data from the tables
  • PCA for all compounds together. Please indicate what were you trying to achieve by doing the compound groups separately (not incorrect, just why)
  • clustering by AHC not just visually

Author Response

Response to the Reviewer comments on manuscript ID: Molecules-1543539

Authors would like to thank for the positive response to our journal submission entitled:  „Effect of Crataegus fruit pre-treatment and preservation methods on extractability of aroma compounds during liqueur production”.  We appreciate the astute observations and constructive comments which helped us to improve our submission. The changes made to the document were highlighted.

Below are our responses to Reviewer’s comments:

In addition to the comments in the pdf, I would urge to authors to address the following issues:

Response:

Revision based on the reviewer’s comments received in the pdf can be found in the manuscript.

Comment:

lack of technical (processing) repeats. As with any product that requires processing, this part can be a big source of variation

Response:

The authors agree that there are possible significant differences between the results obtained during different series of production. In general, different results may be also obtained for plant material derived from different geographical area/stand, during different time or in dependence on weather or soil conditions. Therefore, to minimize variability factors we use one lot of raw material. The aim of the study was to estimate the effect of sugar saturation and method of preservation on the number and concentration of aroma compounds in liqueurs. All analyzed treatments were produced from the same lot of raw material to minimize the effect of other factors (.e.g. place and time of collection, etc.) on the studied features.

The limitation of the study was also provided in the conclusion section of the revised manuscript.

Comment:

how were the compounds identified (there is a footnote for Table 1 that belongs in the M&M section) and how were they quantified? This is also relevant for the stats

Response:

In the present paper the profile of volatile compounds was presented. The substances were identified by mass spectra libraries: NIST08, NIST08s, FF NSC1.3, and by RI â€’ retention index. RI data were retrieved from a reference database of the National Institute of Standards and Technology and compared â€’ based on analyses of n-paraffins â€’ with values calculated  as Linear Retention Indices (LRI)..

The appropriate information was added to the Materials and method section of the revised manuscript (Page 19, lines 4-9)

Comment:

The presentation of the results is exhaustive, please rethink it to make the reader follow the thinking and not simply repeat the data from the tables

Response:

In Table 1 we wanted to show statistically significant differences between particular type of liqueurs in respect of the content of identified volatile compounds with retention times and indexes and their classification into chemical groups and their CAS numbers. On the other hand, Table 2 presents data on the general effect of sugar saturation and/or preservation method (lest square means from the ANOVA) on the shaping of the profile of volatile compounds in the analyzed liqueurs. Moreover compounds in Table 2 were classified into groups and description of the flavors connected with particular compounds was provided according to data found in literature. 

However, if the Reviewer or Editor consider that data in Table 2 presents data supplementary to Table 1, Table 2 can be moved to the Supplementary materials as Table S1.

Also the discussion of the results in the revised manuscript was shortened and given in a more condensed form.

Comment:

PCA for all compounds together. Please indicate what were you trying to achieve by doing the compound groups separately (not incorrect, just why)

Response:

Consideration of all compounds from all classes (alcohols, terpenes, esters etc.) would result in the number of statistical variables that considerably exceeds the number of cases which might very much disrupt the results obtained. Therefore, volatiles were classified into particular groups (classes). This has in a more reliable way reduced the number of variables (presented as two independent components) and allowed to more reliable observation of the no-observable before relations. The aim of the analysis was preliminary investigation which groups/classes can be obtained.

The classes of compounds were provided to give more clear view on the obtained results.

Comment:

clustering by AHC not just visually

Response:

The clusters were established using a hierarchical method (cluster analysis). The selected clusters overlap with the selected groups in most cases, with minor exceptions, like in the case of unsaturated esters, where, apart from clusters (statistical analysis), we wanted to analyse which volatile compounds dominate in which processing methods, which was the main reason for small discrepancies. If the Reviewer wishes, we will send the cluster analysis.

Round 2

Reviewer 2 Report

In my review of the earlier version of this manuscript, I mentioned three areas where I had significant concerns. The authors have made only limited changes to the manuscript itself in response to these but have provided clear and sufficient explanations of their thinking in their response to my comments. I think it would further improve the manuscript, as well as making it more readily understandable for the reader, to have these explanations included in the text.

I have no particular view on whether Table 2 should be in the text or the supplementary information and leave that to the editor.  

I am satisfied that my minor comments have been fully addressed by the authors.

Author Response

Response to the Re-Reviewer comments on manuscript ID: Molecules-1543539

Authors would like to thank for the positive re-response to our journal submission entitled:  „Effect of Crataegus fruit pre-treatment and preservation methods on extractability of aroma compounds during liqueur production”.  We appreciate the astute observations and constructive comments which helped us to improve our submission. The changes made to the document were marked up using Track Changes.

Below are our responses to Reviewer’s comments:

Comment – review 2:

“In my review of the earlier version of this manuscript, I mentioned three areas where I had significant concerns. The authors have made only limited changes to the manuscript itself in response to these but have provided clear and sufficient explanations of their thinking in their response to my comments. I think it would further improve the manuscript, as well as making it more readily understandable for the reader, to have these explanations included in the text.

I have no particular view on whether Table 2 should be in the text or the supplementary information and leave that to the editor.  

I am satisfied that my minor comments have been fully addressed by the authors.”

Response:

As suggested by the Reviewer, respective explanations from our first response to the comments of the Reviewer were included in the revised manuscript (Revision 2), i.e.:

Comment from review 1 (Reviewer 2):

It is difficult to summarise data such as this so as to produce an understandable summary, and the authors have obviously made a lot of effort in this direction. The statistical analysis of the data was very thorough, comprising one-way and two-way analyses of variance intended to investigate the differences between the abundance of individual volatiles between the eight different treatments used; and the overall effects of sugar saturation and preservation respectively. I’m not sure that this really required two separate analyses. Table 2 was very difficult to understand, but it appears to represent the marginal means±SEs of the two-way analyses (this could be stated more clearly). I don’t see why the same overall result could not have been achieved by doing only the two-way analysis and forming linear contrasts between the relevant cells in that design.

The suggested by the reviewer changes were made in the manuscript, i.e.:

Page 3, Lines 115-117: ” Table 1 shows statistically significant differences between particular type of liqueurs in respect of the content of identified volatile compounds with retention times and indexes and their classification into chemical groups and their CAS numbers.”

Pages 10-11, Lines 52-3: “Table S1 presents data on the general effect of sugar saturation and/or preservation method (lest square means from the ANOVA) on the shaping of the profile of volatile compounds in the analyzed liqueurs. Moreover compounds in Table S1 were classified into groups and description of the flavors connected with particular compounds was provided according to data found in literature. “

Comment:

The other arm of the analysis was the use of PCA biplots, which do not seem to contribute very much to the overall story. Scores plots are more commonly used in studies of quality or source attribution, but here the groups are predefined by the design and there are no unknown samples. Whether they cluster or not does not seem add anything to the overall picture presented in Tables 1 and 2. Similarly it is difficult to see what purpose the loadings data serves, unless correlations between volatiles could point to their biochemical origin, though the authors do not discuss this.

The suggested by the reviewer changes were made in the manuscript, i.e.:

Page 12, Lines 41-49: ” The aim of the PCA was additional verification of the obtained results. Multidimensional analysis provides a simple visual representation of the complex analysis of the collected data. This allows to attempt to answer a question e.g. if there are relations between the applied preservation method and volatile compounds present in the sample or which preservation techniques give similar results as regards concentration of volatile compounds. Thanks to the information as to degree of correlation between preservation method and kind and concentration of the volatile compounds it is possible to select the best method of fruit preparation to achieve final product, i.e. liqueur with rich, full-bodied and pleasant flavor.”